# Does Obesity Influence Women’s Decision Making about the Mode of Delivery?

**DOI:** 10.3390/jcm11237234

**Published:** 2022-12-06

**Authors:** Maciej Walędziak, Anna Różańska-Walędziak

**Affiliations:** 1Department of General, Oncological, Metabolic and Thoracic Surgery, Military Institute of Medicine—National Research Institute, Szaserów 128 St., 04-141 Warsaw, Poland; 2Department of Human Physiology and Pathophysiology, Faculty of Medicine, Collegium Medicum, Cardinal Stefan Wyszynski University in Warsaw, 01-938 Warsaw, Poland

**Keywords:** cesarean delivery, obesity, shared decision making, cesarean delivery, maternal request, vaginal delivery

## Abstract

Introduction: The ratio of cesarean deliveries (CDs) has been increasing worldwide, with a growing problem of cesarean delivery on maternal request (CDMR) and an alarmingly increasing rate of CD in the private sector. There are numerous factors influencing women’s preferences for the mode of delivery and their opinion about shared decision making (SDM). Material and method: The study was designed as an online survey, filled in by 1040 women. The questionnaire included questions about women’s preferences for the mode of delivery, their opinions about CDMR and the process of decision making regarding the mode of delivery. Results: There were no statistically significant differences found between women with a BMI ≤ 25 kg/m^2^ and >25 kg/m^2^, nor with a BMI ≤ 30 kg/m^2^ and >30 kg/m^2^, on the subject of the preferred method of delivery, and the opinion regarding SDM and CDMR without medical indications. More than 85% of women in all groups, who preferred CD as the mode of delivery, wanted to have CDMR. Conclusion: We have not found obesity and overweight to be a factor influencing women’s preferred mode of delivery, their opinion about SDM and their preference for CDMR without medical indications. However, the sample size of women with morbid obesity with a BMI ≥ 35 kg/m^2^ was too small for the results to be considered significant in this group, and it will therefore be subject to further studies.

## 1. Introduction

The global cesarean delivery (CD) ratio has increased more than two fold over the last two decades and increases every year by 4%. According to the World Health Organization (WHO), the worldwide CD ratio has reached 21% in 2018, with a wide span from 5% in southern African countries to almost 60% in some South American countries. About 6.2 million CDs are estimated to be annually performed unnecessarily worldwide, the leaders being Brazil and China [1]. Following the WHO and the Pan American Health Organization (PAHO), it is to be stated that “there is no justification for any region to have a caesarean section rate higher than 10–15%” [2]. CDs performed without a background of well-prepared multi-specialists and adequate healthcare facilities, and in particular including a high standard of neonatal care, can lead to the harm or even death of women and child. An age of 40 years old or more, obesity and a history of CD are often considered, even by experienced obstetricians, as indications for a CD, which should not happen following the international recommendations for CD. Additionally, national recommendations for CD are not consistent and often allow women the liberty to decide their mode of delivery, disregarding the absence of the presence of medical indications. Due to the increasing number of CDs, pressure has started to rise on the obstetricians in some countries which can sometimes lead to unnecessary, risky attempts at VD [3].

There are not only differences in the approach to cesarean delivery on maternal request (CDMR) between countries; there are also differences in the percentages of CDMR being performed between the public and private sectors. In the countries where the role of the private sector is important in pregnancy care, including the United States, Brazil, India or Iran, the differences are visible both in the approach of obstetricians and in the women’s preference for CDMR. Almost 32% of women in India, and more than 20% in Iran, deliver in private hospitals [4,5]. In some countries, such as Thailand, the rate of cesarean section in private care is almost ten-fold higher than in the public sector. In others, such as Rwanda, it is four times higher, and in India it is three times higher [6,7]. The situation is even more alarming in Bangladesh, where among the 50% of deliveries that take place in medical facilities, almost two-thirds are in private care facilities, where the CS ratio is 83% [8]. Brazil is known to have one of the highest rate of CS in the world, with the CS ratio over 80% in private care hospitals in central-west Brazil and more than 95% in hospitals in southern Brazil [9,10]. One of the reasons for that situation, mentioned by the Brazilian authors, is the problem of ‘defensive medicine’ and CS performed by obstetricians who want to avoid lawsuits [11].

The increased rate of CS in the private sector, in contrast to the public sector, is also associated with a higher prevalence of CS in higher socioeconomic status groups. In some countries, including India, the public sector also pays for CSs, and therefore the rate of CSs in public sector is strongly influenced by the percentage of women from different socioeconomic status groups [12]. There are some additional factors that can explain the differences in the CS rate between private and public sector; for example, there is a tendency for women who use private medical care and are of an older age to consider a CS as a better way to avoid birth complications [13].

The concept of shared decision making (SDM) has been known for over fifty years. SDM means engaging patients in making decisions about the diagnostic process, treatment and follow-up [14]. SDM has been described as the optimum approach to making decisions about patient care in clinical situations, especially where there is uncertainty about the choice of different treatment options [15]. There are special tools created to facilitate the understanding of the health implications of different options of treatment for the patients [16,17]. There are a lot of research data suggesting that inadequate information regarding different modes of delivery and individual medical circumstances may cause a number of complications, including post-traumatic stress disorder symptoms [18]. SDM gives patients knowledge about the procedure, reduces the level of anxiety and improves the results of treatment. SDM is of utmost importance in the case of deciding about the mode of delivery, as there are two patients involved in the choice, the mother and the baby. There is a necessity for women to be well informed about the risks and benefits of the different modes of delivery, and this education should be conducted by the health care professionals [19].

Weight management has become an important current problem, given the ratio of obese patients having reached 40% in the US [20]. The risk of complications during delivery is about three-fold higher in obese women than in a population with a normal body weight [21,22]. Obesity also increases the risk of precipitated CD due to intra-labor indications, especially those which are cardiotocography (CTG)-related [23]. There is also a higher risk of surgical site infection after CD in obese women [24].

The most important issue is finding the optimum balance between the risks and benefits when considering the mode of delivery for each pregnant woman, to make the delivery as safe as possible for both the woman and the child. In the beginning, it is necessary to understand the factors influencing the decision making.

The aim of this study was to evaluate whether obesity had any influence on women’s opinions regarding the mode of delivery, CDMR and their preference for decision making.

## 2. Materials and Methods

The study was designed as an anonymous online survey and was performed in 2020. A total of 1040 women filled in the questionnaire that had been distributed via social media. Women aged 18 and older were invited to take part in the survey, which included questions about their basic characteristics (age, place of residence, education, socioeconomic status, height, weight, comorbidities and obstetric history). The main part of the questionnaire included questions about women’s preference for the mode of delivery, their opinion about CDMR and the process of decision making regarding the mode of delivery.

There were no exclusion criteria apart from gender, age less than 18 years old and missing or conflicting data. Questionnaires including incorrect data were excluded from further analysis. The materials and methods for this study were also described in previous publications [25,26].

### 2.1. Statistical Analysis

Statistical analysis was performed using Statistica 13 (StatSoft. Inc., Tulsa, OK, USA). The Mann–Whitney U test and Student’s *t*-tests were used for quantitative data comparison as required. A two-sided Fisher’s exact test and chi-square test were used for categorical and binary data comparison as required. A *p* value < 0.05 was considered significant.

### 2.2. Ethical Considerations

The study was anonymous, and was performed in accordance with the ethical standards laid down in the 1964 Declaration of Helsinki and its latter amendments (Fortaleza). Participants were informed about the aim of the study and informed consent was obtained electronically prior to the beginning of the survey. The approval from the Warsaw Medical University Ethics Committee was obtained on 19 March 2013, the code AKBE/21/13.

## 3. Results

### 3.1. Characteristics of the Study Group 

Most of the respondents were of childbearing age (defined as aged between 18 and 45 years old), and the mean age was 32.0 (±6.7). The mean BMI was 24.43 (±5.02). More than 50% of the women lived in cities with over 100,000 inhabitants. The majority of women declared higher education, 683 (63.65%), and medium socioeconomic status, 792 (73.95%). A total of 71.15% of the participants denied having any chronic diseases, 35 women (3.44%) had hypertension and 28 women (2.75%) had diabetes mellitus type 2. A total of 9.08% respondents were pregnant during the filling out of the survey. 

The baseline characteristics of the group are presented in Table 1.

### 3.2. General Opinion about Optimum Mode of Delivery

A total of 589 respondents (56.6%) declared their preference for VD, 252 (24.2%) declared a preference for CD and 199 (19.1%) had no opinion. A total of 359 (34.5%) women wanted to independently decide their method of delivery, 463 (44.5%) chose SDM with their obstetrician, 204 (19.6%) wanted CD only for medical indications and 14 (1.3%) had no opinion. A total of 472 (43.4%) women would decide to have a CDMR without medical indications and 357 (34.3%) were against the idea, with 211 (20.3%) stating no opinion.

### 3.3. Division of the Study Population into Subgroups

For the purposes of the study, we divided patients into groups according to their BMI. The first division included two groups: patients with a BMI 25 kg/m^2^ and lower (63.4% of women) and those with a BMI of more than 25 kg/m^2^ (36.6%). The next division included women with a BMI ≤ 30 kg/m^2^ and >30 kg/m^2^, or 87.6% and 12.4% of the participants, respectively. Finally, we compared the four subgroups of patients: BMI ≤ 25 kg/m^2^ (63.4%), 25–30 kg/m^2^ (24.2%), 30–35 kg/m^2^ (8.9%) and ≥35 kg/m^2^ (3.5%).

### 3.4. Women with Normal or Low BMI vs. Overweight and Obese Women 

There were no statistically significant differences found between women with a normal or low BMI and those with a BMI higher than 25 kg/m^2^ on the subject of their preferred method of delivery, and their opinion about SDM and CDMR without medical indications. CD was the preferred mode of delivery of 24.0% of women with a BMI ≤ 25 kg/m^2^ and 24.7% of women with a BMI higher than 25 kg/m^2^. In the group with a BMI ≤ 25 kg/m^2^, 32.9% wanted to decide independently about the mode of delivery, 44.5% wanted to undergo SDM with their obstetrician and 21.2% declared they would prefer CD only due to medical indications. Out of overweight and obese women, 37.2% preferred self-deciding, 44.6% preferred SDM and 16.8% preferred CD only for medical indications. There was a lower rate of preference for CD only for medical indications in the overweight and obese group, but the difference was not statistically significant. A total of 34.6% of women with normal and low body weight declared they would prefer to have CDMR, with 44.9% against it, compared to 33.9% vs. 46.1% of women with a BMI of more than 25 kg/m^2^.

### 3.5. Obese and Non-Obese Women

The results were similar for the groups of women with a BMI ≤ 30 kg/m^2^ and >30 kg/m^2^. A total of 24.1% of women with a BMI ≤ 30 kg/m^2^ and 24.8% of obese women preferred CD. Furthermore, 34.5% of non-obese women indicated a preference for self-deciding their mode of delivery, 44.3% stated a preference for SDM and 19.9% stated a preference for CD for medical indications only, with 34.1%, 45.7% and 17.8%, respectively, in the obese group. A total of 34.1% of obese women would decide to have a CDMR, with 45.0% against it, compared to 34.3% vs. 45.4% in the non-obese group.

### 3.6. Comparison between Four Subgroups of Women

No differences were found in the four subgroups—BMI ≤ 25 kg/m^2^, 25–30 kg/m^2^, 30–35 kg/m^2^ and ≥35 kg/m^2^—in regards to the preferred method of delivery, opinion about SDM and preference for CDMR without medical indications.

In the group of women with a BMI 25 kg/m^2^ and lower, 29.0% of women who preferred VD wanted to self-decide their mode of delivery vs. 71.0% who preferred SDM; among those who preferred CD, 65.2% chose self-deciding vs. 34.8% SDM (*p* < 0.05). The results were comparable in the group of overweight and obese: 30.1% of those who preferred VD would like to independently decide their mode of delivery vs. 69.9% who favored SDM, with 72.5% vs. 27.5%, respectively, for those who preferred CD (*p* < 0.05). Similar results were observed in the groups of women with a BMI ≤ 30 kg/m^2^ and >30 kg/m^2^.

86.1% of women with a BMI ≤ 25 kg/m^2^ who preferred CD would like to have a CD without medical indications, compared to 10.5% of women who preferred VD (*p* < 0.05). In the group of overweight and obese women, 85.5% of women with a preference for CD declared they would have CDMR, with only 11.1% favoring VD (*p* < 0.05). In the group of women with a BMI ≤ 30 kg/m^2^, 86.2% of those who preferred CD wanted CDMR compared to 10.3% of those who preferred VD (*p* < 0.05).

Among women with a BMI ≤ 25 kg/m^2^ who preferred self-deciding their mode of delivery, 67.7% would like to have CDMR, compared to 27.0% of those who chose SDM (*p* < 0.05). A total of 62.7% of women with a BMI > 25 kg/m^2^ who wanted to self-decide would prefer to have CDMR compared to 22.4% of those who chose SDM (*p* < 0.05). The results were comparable for the groups of women with a BMI ≤ 30 kg/m^2^ and >30 kg/m^2^.

The results in the group of women with a BMI ≥ 35 kg/m^2^ were not statistically significant because of the small sample size of 35 participants.

## 4. Discussion

Although the idea of SDM has been known for decades, the increase in patient awareness that has progressed in recent years has redirected the problem of choosing the optimum mode of delivery to a team consisting of the obstetrician and the pregnant woman [27]. This often creates a difficult issue, as women are often influenced by their beliefs, families, friends and social media and do not accept the true advantages and disadvantages of different modes of delivery for themselves and their babies [28]. The woman’s belief in the superiority of caesarean section often has no medical justification. The decision about the mode of delivery involves the health of the baby and therefore is not simply a case of a woman’s decision and preference, as the decision involves the health and well-being of two people, one of whom cannot decide for themself [29]. A top priority is the well-prepared education of pregnant women in the different modes of delivery, which is emphasized by many national societies of gynecologists and obstetricians, including The Canadian Society of Gynaecologists and Obstetricians [30,31]. 

Our results indicate that the preference for CD as the mode of delivery is in correlation with a woman’s need to self-decide. The vast majority of women who preferred CD to VD wanted to have CDMR without medical indications. The preference for CD is common in women who prefer to have control over every aspect of their life [32]. They think about CD as an opportunity to schedule their delivery, which otherwise would come unexpectedly. According to a recent systematic review by Coates et al., there are six main reasons associated with birth preference: the perception of safety, the fear of pain, previous birth experience, encouragement from health professionals, social and cultural influences and access to information and educational levels [33]. In a study by Reyes et al., fear was reported as the main factor that influenced their decision by 73.2% women who reported a preference for CD [34].

The factors influencing women’s preferences for the mode of delivery differ between countries, and are dependent upon social and cultural beliefs, country income and level of education. Preis et al. found that in the Israelite population the main reasons for a preference for CD were a history of previous VD, being less religious, believing that birth was a medical process and a negative experience [35]. In Norway, CD was preferred by women with symptoms of depression, an age over 35 years old, a history of previous CD, prolonged labor or a negative birth experience [36,37,38]. In China, the preference for CD was indicated especially for the possibility of choosing a lucky day for the baby’s birth, followed by an age of 40 years old and above, ethnic minorities, difficulties in becoming pregnant, and the husband’s preference. Chinese women who preferred CD believed in the higher safety of CD vs. VD, the lower level of pain and thought that CD was a better choice for the baby’s and the woman’s health [39]. In a study by Hatamleh et al. from Jordan, the main reasons for CDMR were a fear of VD, concerns about future sexual life, the need for humanized birth and the decision-making process [40]. In an Iranian study by Rajabi et al., the main factors for women’s preference for CD were age, spousal educational level, the number of live births and preconceived attitudes about delivery [41]. In Ethiopia, the factors influencing women’s preferences for CD were mostly previous or present pregnancy complications and a lack of access to cardiotocography during delivery [42]. 

Although in our study obesity was not established to be a reason for choosing CDMR, there are studies that reported maternal obesity as one of the reasons for asking for CDMR [43,44]. Zhou et al. analyzed a group of over 1,000,000 Chinese women, who had almost 94,000 CDMRs. Overweight was defined as a BMI 23 to <27.5 kg/m^2^ and obesity as BMI ≥ 27.5 kg/m^2^, according to the WHO recommendations for the Asian population. Maternal obesity was associated with an increased ratio of overall CDs, as well as CDMR, with the growing ratio correlated with the BMI index [45].

Obesity is a well-established risk factor for maternal delivery complications. It is associated with failure in the induction of labor, the necessity of using multiple induction agents and higher doses, prolonged labor and an increased risk of CD with the increasing BMI. Obesity increases the risk of labor dystocia and obstetric anal sphincter injuries. In a study by Drummond et al., the risk of intra-labor CD in nulliparous women was 18.3% in the group with a BMI < 30 kg/m^2^ and 48.2% in the group with a BMI ≥ 40 kg/m^2^ [46]. Additionally, obesity is associated with a higher risk of post-term pregnancies [47]. Obesity, especially with a BMI ≥ 40 kg/m^2^, is also an independent risk factor for surgical wound complications after CD [48]. Additionally, it increases the risk of hemorrhage, blood transfusion, unplanned hysterectomy, longer postpartum hospitalization and intensive unit care admission [49,50].

Obesity is also a considered risk factor for overweight and obesity in infants; however, the reported results can be confounding. Li et al. performed a meta-analysis to examine whether cesarean section increased the risk of later overweight and obesity. They found that cesarean section was moderately associated with offspring overweight and obesity [51]. CDMR can have an additional influence on the future body weight and development of children. In a recent study by Zhou et al., elective CD was found to have an association with central obesity and hypertension in children aged 4 to 7 years old [52]. In a study by Li et al., CDMR was found to be correlated with the lowest probability of developing psychopathological problems in children of preschool age, followed by spontaneous vaginal delivery and associated vaginal delivery [53].

### Limitations of the Study

The anonymous survey was conducted online, and the participants were therefore limited to women who had the access and ability to use a computer and social media. This form of distribution also excluded the possibility of the direct control of the respondents or the calculation of the response rate, although there was no incentive to introduce dishonesty into responses. Additionally, another possible limitation to our study may be recall bias, considering the obstetric history of the patient and the subjectivity of the patient’s responses.

## 5. Conclusions

Women’s preferences for their mode of delivery result from numerous factors. In our study, we have not found obesity and overweight to be a factor influencing women’s preference for the mode of delivery, their opinion about SDM and their willingness for CDMR without medical indications. However, the sample size of women with morbid obesity with a BMI ≥ 35 kg/m^2^ was too small for the results to be considered significant in this group, and it will therefore be subject to further studies.

## Figures and Tables

**Table 1 jcm-11-07234-t001:** Baseline characteristics of the study group.

Variable	2020
Respondents [n]	1033
Mean age [years] (SD)	32.0 (±6.7)
Mean BMI [kg/m^2^] (SD)	24.43 (±5.02)
Place of habitation:	
Cities > 100 000	613 (57.08%)
Cities 50 000–100 000	120 (11.17%)
Towns < 50 000	146 (13.59%)
Villages	195 (18.16%)
Education:	
Primary	19 (1.77%)
Secondary	283 (26.37%)
Higher	683 (63.65%)
Medical	88 (8.20%)
Socioeconomical status:	
Low	15 (1.40%)
Medium	792 (73.95%)
High	264(24.65%)
No comorbidities	725 (71.15%)
Hypertension	35 (3.44%)
Diabetes mellitus type 2	28 (2.75%)
Ongoing pregnancy	
Yes	97 (9.03%)
No	977 (90.97%)

## Data Availability

The data presented in this study are available on request from the corresponding author. The data are not publicly available.

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
