# Peer review of "Does Obesity Influence Women’s Decision Making about the Mode of Delivery?"

_jcm, 2022, doi:10.3390/jcm11237234_

Round 1
Reviewer 1 Report
1 Discussion must be more long.
2 Some references are necessary. The impact of cesarean section on offspring overweight and obesity: a systematic review and meta-analysis
3 Result must be clearly wrote. The result need of order in paragraph. If possible a tablet of results
4 Grammar need to be revised
Author Response
Dear Reviewer 1
Thank you for your remarks. We corrected the manuscript accordingly.
Following your remark, we added the suggested references and analyzed them in the Discussion.
According to your suggestion, we corrected the Results section and divided it into separate paragraphs. We also added tables presenting the results.
Reviewer 2 Report
The Authors of this article tired to answer a very interesting question. However in my opinion they failed.
In the introduction the cited articles have no differentiation in percentage of performing CS on request in public or private sector. Generally the number of CS is growing but there is no reliable data on this topic.
An online survey even on over a 1000 participants is still a anonymous survey.
The only argument with which I am able to agree is the global education of patients and their families
If the Authors would consider this I suggest to cooperate in writing with an Ob&Gyn specialist.
Author Response
Dear Reviewer 2,
Thank you for your evaluation of our manuscript.
We sincerely hoped our study would give an answer we were interested in getting, but there is still need to analyze the problem further and on a larger group of patients.
In our country, the private sector is responsible for less then 2% of all deliveries – the private delivery care is marginal and therefore we did not underline the differences existing in other countries between private and public sectors. We absolutely agree that available data about the growing number of cesarean deliveries is not reliable, as we can observe, even in our country, a tendency to lower the presented rate of cesarean sections to better fit the recommendations of international and our national society of obstetricians.
Following your remark, we added information that the survey was anonymous and the information about possible bias is also included in the manuscript.
The problem of education of patients themselves, not even mentioning their families is very strong in our country. Obstetric care is based mostly on obstetricians, the role of midwives is unfortunately negligible during pregnancy and less than 30% pregnant women go to pre-delivery schools even in cities of more than 100,000 inhabitants.
As to the cooperation with an Ob-Gyn, the team leader, who created the project of the manuscript is an Ob-Gyn specialist. This study is a part of a larger project about women’s opinion about the optimum mode of delivery, shared decision-making, cesarean section on woman’s request and analgesia during delivery.
Round 2
Reviewer 2 Report
The Authors did not respond to reviewer comments properly, particularly in describing the private section in the Introduction. Moreover, the flow of the manuscript is confusing, with too many tables. Therefore, it's very difficult to get the message of the manuscript. No opinion of the Ob&Gyn Specialists is added.
Thank you for the opportunity to consider your work. I am sorry that I cannot be more positive on this occasion.
Author Response
Dear Reviewer 2,
Please excuse the misunderstanding about the introduction. It was rewritten now, with an analysis of the differences between the public and private sector of medical care, hopefully addressing your remarks in a proper way.
The presentation of the results section and division into paragraphs was made because of request of the other Reviewer, who also required the additional tables that appeared in the new version of the manuscript.
I am very sorry, but I do not exactly understand the idea of the Ob&Gyn specialist opinion that you ask for, as I mentioned in the previous reply that an Ob&Gyn had already been part of the research team.
Anna RóżaÅ„ska-WalÄ™dziak is an Ob&Gyn specialist and an expert in the field of pregnancy after bariatric surgery. She authorized the whole project about women’s preference for the mode of delivery, opinion about shared-decision making and cesarean delivery on demand. Dr Anna RóżaÅ„ska-WalÄ™dziak positively recommended the subject of the present study and participated in the creation of the manuscript.
Of course I understand that the results that we obtained are not substantial and we hoped for more specific ones, but we are planning to study the subject further. Thank you for your understanding.